# Dual RNA-Seq Analysis Reveals Transcriptome Effects during the Salmon–Louse Interaction in Fish Immunized with Three Lice Vaccines

**DOI:** 10.3390/vaccines10111875

**Published:** 2022-11-07

**Authors:** Antonio Casuso, Valentina Valenzuela-Muñoz, Cristian Gallardo-Escárate

**Affiliations:** 1Interdisciplinary Center for Aquaculture Research (INCAR), Universidad de Concepción, Concepción 4030000, Chile; 2Laboratory of Biotechnology and Aquatic Genomics, Department of Oceanography, Universidad de Concepción, Concepción 4030000, Chile

**Keywords:** vaccine, *Caligus rogercresseyi*, *Salmo salar*, transcriptome, dual RNA-seq, ectoparasite, host-parasite interaction

## Abstract

Due to the reduced efficacy of delousing drugs used for sea lice control in salmon aquaculture, fish vaccines have emerged as one of the most sustainable strategies in animal health. Herein, the availability of *C. rogercresseyi* and *Salmo salar* genomes increases the capability of identifying new candidate antigens for lice vaccines using RNA sequencing and computational tools. This study aimed to evaluate the effects of two recombinant antigens characterized as peritrophin and cathepsin proteins on the transcriptome profiling of Atlantic salmon during a sea lice infestation. Four experimental groups were used: Peritrophin, cathepsin, and peritrophin/cathepsin (P/C), and PBS as the control. *C. rogercresseyi* female, *S. salar* head kidney, and skin tissue samples were sampled at 25 days post-infestation (dpi) for Illumina sequencing and RNA-seq analysis. Differential gene expression, gene ontology, and chromosomal expression analyses were performed. Furthermore, the dual RNA-seq analysis approach was performed to simultaneously explore host and pathogen transcriptomes, identifying functional associations for vaccine design. The morphometry of female sea lice exposed to immunized fish was also evaluated. The RNA-Seq analysis exhibited prototype-dependent transcriptome modulation, showing a conspicuous competition for metal ions during the infestation. Moreover, Dual RNA-seq analysis revealed vaccine-dependent gene patterns in both the host and the pathogen. Notably, significant morphometric differences between lice collected from immunized and control fish were observed, where cathepsin and P/C showed 57% efficacy. This study showed the potential of two proteins as lice vaccines for the salmon industry, suggesting novel molecular mechanisms between host–parasite interactions.

## 1. Introduction

As a food source, aquaculture production has increased the need to develop sustainable alternatives to control diseases, such as vaccines [1,2]. Vaccines have been used widely and effectively in salmon farming to control viral and bacterial diseases [2,3,4]. However, developing vaccines to control parasitic diseases represents one of the biggest challenges for the salmon industry [4]. One of the most relevant diseases in salmon aquaculture is provoked by sea lice infestations. These copepod ectoparasites generate high economic losses for the salmon farming industry [5,6,7]. The main species are *Lepeophtheirus salmonis*, which is predominant in Norway and Canada [8,9], and *Caligus rogercresseyi* in Chile [5,10,11]. These ectoparasites feed on the mucus and blood of their host and trigger skin lesions, weight loss, and immunosuppression [7,12]. Currently, the primary control method used is pharmacological treatments, which have been losing efficacy over time [13,14]. There are also thermal and mechanical treatments, such as spraying the fish with water under pressure or submitting them to turbulence, which generates fish stress [15,16].

Different research groups have explored the use of vaccines for controlling sea lice infestations [17,18,19,20,21,22]. Among them, the effect of proteins such as potassium chloride transporter [19], amino acid transporter (P33) [19], peptide my32 [18], and chimeric proteins TT-P0 [21] have potentially been reported as immunogenic molecules. These investigations have described inflammatory and antioxidant responses that, in turn, can modulate the host immune response in the fish hosts [18,21]. Another example is the vaccine called IPath, a chimeric protein design using the subunit H of ferritin and transferrin genes of the Atlantic salmon [17]. The use of IPath^®^ reported an adult *Caligus* reduction of more than 90%. In addition, it shows an effect on the fish ferritin transcripts modulation [17]. Furthermore, alterations in the transcriptome profiles of *C. rogercresseyi* have also been reported; there was negative regulation of vitellogenin and protease genes [23]. The effects on the female phenotype exposed to immunized fish were also observed, such as abdominal inflammation and alteration in the egg strings [23].

During sea lice infestation, the fish host displays different molecular mechanisms to cope with the parasitism. For instance, genes related to pro-inflammatory pathways, specifically the TLR22 [24], and the nutrition immunity strategy [25] have been observed. In parallel, the sea louse evades the host response by increasing the expression of antioxidant genes or iron transport-related genes such as ferritin [26]. Notably, the life cycle of *C. rogercresseyi* has been deeply studied through the transcriptomes during the ontogeny [27]. This relevant molecular knowledge was improved with the recent publication of the first genome assembly at the chromosome level of this species, allowing a better understanding of host–parasite interactions [28]. Collectively, genomic investigations in sea lice can also be applied to discover immunogenic proteins with potential use in vaccine formulations. Herein, proteins associated with the *C. rogercresseyi* secretome, such as cathepsin, and proteins associated with sea lice feeding, such as Peritrophins, emerge as putative antigens [29].

Until now, vaccine effects have been evaluated mainly from the host perspective. However, host–parasite interplays can be affected by the immunization of specific proteins with functional roles during the life cycle of sea lice. Herein, a dual RNA-seq approach can be applied to understand and identify, in parallel, the molecular effects of vaccine formulation in both parasite and fish hosts. Interestingly, a recent study by our research group reported a sea lice burden reduction and transcriptional changes in Atlantic salmon immunized with recombinant Cathepsin and Peritrophins seven days after sea lice infestation [29]. This study aimed to explore the effect of Atlantic salmon immunized with Cathepsin and Peritrophin antigens and *Caligus rogercresseyi* collected after 25 days post-infestation, evaluating the transcriptome profiling of host–parasite interactions and the phenotypic effects of lice exposed to vaccinated fish. Collectively, the reported study revealed a potential application of the formulated vaccine based on the disruption of the sea louse–Atlantic salmon interaction, producing novel scientific knowledge for salmon-farming aquaculture.

## 2. Materials and Methods

### 2.1. Fish Vaccination and Challenge Trial

The study was conducted according to the guidelines of the 3R and approved by the Ethics, Bioethics, and Biosafety Committee of the Vice-Rectory for Research and Development of the University of Concepción, Chile. Four prototype vaccine formulations were made. Recombinant proteins of peritrophin, cathepsin, and a combination of peritrophin and cathepsin (50%/50%) (P/C) were used as antigens and PBS as the control group. Atlantic salmon of 120 g were acclimated for two weeks in the experimental laboratory of the Marine Biological Station, University of Concepción, Dichato, Chile. Fish were injected intraperitoneally in triplicate and divided into four experimental groups with 20 fish per tank, according to the vaccine formulation: Peritrophin, cathepsin, P/C, and PBS. Each vaccine prototype was formed with 30 µg of the antigen in a dose of 100 µL, along with the adjuvant (MONTANIDETM ISA 761VG) in 30/70 *v/v* proportions (antigen/adjuvant). Atlantic salmon were injected with 100 μL of the formulations. Then, 400 accumulated thermal units (ATUs) were infested with 35 copepodites per fish. At 25 days post-infestation (dpi), sea lice burden was registered. The efficacy of each lice vaccine formulation was evaluated using the Shapiro–Wilk test for statistical analysis for data distribution and then tested by one-way ANOVA and Tukey posthoc analysis for significant differences in parasite burden. The non-parametric Kruskal–Wallis test was used to analyze the number of eggs per female. Significant statistical values occur at *p* < 0.05. In addition, female sea lice, fish skin, and head kidney tissue samples were collected, fixed in RNA Stabilization Reagent^®^ (Ambion, Life Technologies ™, Carlsbad, CA, USA), and stored at −80 °C until RNA extraction.

### 2.2. RNA Extraction and High-Throughput Transcriptome Sequencing

Total RNA was extracted from Atlantic salmon skin and head kidney tissues of five individuals per experimental group, considering three replicates per fish group. In addition, ten sea lice per replicate and the vaccinated group were pooled for RNA isolation (n = 30 individuals per treatment). Total RNA extraction was performed using the Trizol Reagent (Ambion, Carlsbad, CA, USA), following the manufacturer’s instructions. The isolated RNA’s quality, purity, and quantity were measured in TapeStation 2200 (Agilent Technologies Inc., Santa Clara, CA, USA). The pooled sea lice and Atlantic salmon samples collected from each experimental group were subsequently used for TrueSeq Stranded mRNA Illumina library synthesis. NGS sequencing was performed by the Macrogen Inc. (Seoul, Korea) company on the HiSeq Illumina platform. Raw reads were trimmed using the Trim Reads plug-in included in the CLC Genomics Workbench (Version 21.0.3, Biomatters Ltd.a., Auckland, NI, NZ). De novo assemblies were performed using datasets from each tissue and treatment prototype, with a contig length > 200 bp, a mismatch cost of 2, insert and deletion costs of 3, a length fraction of 0.8, and a similarity of 0.9. Contigs were annotated by BlastX analysis using a database constructed from GenBank and UniprotKB/Swiss-Prot, with an E-value cutoff value of 1 × 10^−5^.

### 2.3. RNA-Seq and Gene Ontology Analyses

The transcriptome databases of *S. salar* and *C. rogercresseyi* from experimental groups were used for RNA-seq analyses. The reads were mapped separately against all annotated contigs with the CLC Genomic Workbench software. The settings were minimum length fraction = 0.8 and minimum similarity fraction (long reads) = 0.8; the expression value was established as transcripts per million reads (TPM). The distance metric was calculated with the Manhattan method, and Kal’s test was used to compare gene expression levels in fold change (*P* = 0.0005, FDR corrected). Differential expression analysis was conducted for tissue samples and filtered by absolute fold-change values ≥ 4 and *p*-value < 0.05. Contigs were annotated by BlastX using UniProtKB/Swiss-prot databases. The BLOSUM62 matrix was used with a cutoff E-value of 1E^−10^. Gene Ontology (GO) enrichment analyses were performed to identify the most-represented biological processes (BP) and molecular functions (MF) of unique contigs of each vaccinated group (differentially expressed). The Blast2GO plug-in of the CLC Genome software was used with predetermined parameters.

### 2.4. RT-qPCR

Expression validation using RT-qPCR was performed for seven selected genes of *S. salar* and *C. rogercresseyi* (Appendix A). The same RNA pool used for the high-throughput sequencing was used for RT-qPCR. The RT-qPCR standardization was carried out according to the MIQE guidelines [30]. cDNA was synthesized with 200 ng/µL of total RNA and using the RevertAid H Minus First Strand cDNA Synthesis kit (Thermo Fisher Scientific, Waltham, MA USA), following the manufacturer’s directions. The RT-qPCR reaction was performed on the QuantStudio^TM^ 3 Real-Time PCR System (Applied Biosystems, Life Technologies, Foster City, CA, USA). The comparative ΔΔCt method was used for gene expression quantification [31]. The selection of the reference gene for the experiment was based on evaluating the stability of the elongation factor-α, b-tubulin, and 18S genes by NormFinder. Through this, the elongation factor-α and b-tubulin were selected as reference genes for *S. salar* and *C. rogercresseyi*, respectively. The PowerUp^TM^ SYBR^TM^ Green Master Mix (Thermo Fisher Scientific, Waltham, MA USA) was used in a final reaction volume of 10 μL. Amplification was carried out under the following conditions: 95 °C for 10 min, 40 cycles of 95 °C for 15 s, and alignment temperature for 30 s (Appendix A), followed by 30 s at 72 °C. Each reaction was carried out in a final volume of 10 μL. Statistical analyses were performed in the GraphPad Prism 6.0 software (San Diego, CA, USA). The Shapiro–Wilk test for statistical analysis determined the distribution of data. Additionally, the data were evaluated by one-way ANOVA and Tukey’s post hoc analysis determined significant differences. Statistically significant values were set at *p* < 0.05.

### 2.5. Chromosome Gene Expression Analysis

The Chromosomal Gene Expression (CGE) index was used to evaluate whole-genome modulation according to Valenzuela-Muñoz, et al. [32]. Briefly, raw data from each vaccine candidate were mapped to the *S. salar* genome (NCBI RefSeq assembly accession: GCF_905237065.1). Raw data from lice recovered from vaccinated fish were mapped using the genome of *C. rogercresseyi* (NCBI access: PRJNA551027) [28]. The CGE index represents the percentage of the transcriptional variation between the four experimental groups. For this, we identified the mean coverage of the mapped transcripts in a specific chromosomal region and compared it between experimental conditions. A threshold of 2000 to 100,000 reads in a 5-position window size was used to calculate the transcript coverage value. Threshold values were calculated for each chromosome region using the Graph Threshold Areas tool in CLC Genomics Workbench v21 software. Chromosomes with CGE index values > 60% were visualized with the transcript coverage for each dataset in the Circos software [33]. Subsequently, the genes associated with the CGE = 100% regions were identified. Differential expression analysis was performed on these genes, compared with the PBS control group.

### 2.6. Dual RNA-Seq Analysis Using Gene Expression Correlations

Host–parasite interplay was inferred by gene expression correlation among differential expressed genes of *S. salar* and *C. rogercresseyi*. TPM expression values were used to determine the Pearson correlation coefficient (r) between the genes of *S. salar* and *C. rogercresseyi*. The correlation analysis was performed in R and visualized in RStudio using the Corrplot package [34]. Correlation transcript expression between *S. salar* and *C. rogercresseyi* genes was statistically significant if the r value was >0.9.

### 2.7. C. rogercresseyi Morphometry Analysis and Vaccine Efficacy

Geometric morphometry analyses were conducted in *C. rogercresseyi* adults collected from Atlantic salmon immunized with peritrophin, cathepsin, P/C, and PBS. Body dorsal-side photographs of ten male and female individuals were taken using a Motic model SMZ-171 stereomicroscope connected to a 5 MP Moticam camera (2× objective). Length and width were determined with the Motic Images Plus 3.0 program. Geometric morphometry with landmark analysis was used to determine the effect of lice exposure on fish immunized with the vaccine prototypes. Twenty-nine landmarks located in the contour of the lice were used. Landmark digitization was performed in TpsDig [35]. Digitization error and data processing were estimated in Morpho J [36]. Differences between treatments were determined by analyzing canonical variables using Procrustes distance with 10,000 permutations [37,38]. Vaccine efficacy was calculated concerning the parasite burden of the PBS group, according to Casuso et al. [29].

## 3. Results

### 3.1. Transcriptome Modulation in Vaccinated Atlantic Salmon

RNA-seq analyses were used to assess the transcriptome modulation of immunized fish. Moreover, the observed expression was validated through the relative expression analysis (Appendix A). RNA-Seq analysis of immunized Atlantic salmon’s head kidney and skin tissue showed different transcript patterns among experimental groups (Figure 1A). For instance, in head kidney tissue, the transcripts associated with salmon immunized were separated from the control group. Notably, a cluster of transcripts highly expressed in the head kidney of the control group was downregulated in the P/C group (Figure 1A). In addition, the head kidney tissue from fish immunized with cathepsin exhibited the highest number of transcripts (3436) differently modulated (Figure 1B,C). For the skin tissue, the heatmap representation showed a clustering among skin samples obtained from the cathepsin, peritrophin, and control groups, while the P/C group clustered separately. Interestingly, at difference with the head kidney tissue in skin tissue, a high number of transcripts was observed differently modulated in samples obtained from fish immunized with peritrophin protein (Figure 1B–D).

The GO annotation of genes differentially expressed in each experimental group showed a high number of genes associated with the immune response. For instance, the head kidney tissue of fish immunized with peritrophin exhibits a high number of genes associated with an innate immune response, I-kappa B kinase/NK-kappa B signaling (Appendix A). Furthermore, in the head kidney of fish immunized with P/C, biological processes were associated with an immune response, such as cytokine-mediate signaling pathways and adaptative immune response regulation of defense (Appendix A).

### 3.2. Chromosome Expression of Immunized Salmo Salar

Transcriptome differences among the experimental groups were evaluated at the chromosome level using an index denoted as Chromosome Gene Expression (CGE) (Figure 2A,B) [32]. This index allows for determining the expression differences among experimental groups in a complex dataset. The CGE index is calculated using the mean coverage of transcripts mapped onto a specific chromosome region and compared among experimental conditions. Here, the CGE index estimation suggests a significant effect in Atlantic salmon skin triggered by the immunization (Figure 1B), with a high number of chromosome regions with CGE values over 60%, which indicates high expression differences among the experimental groups in some regions of the *S. salar* genome. Notably, in skin tissue, 25 chromosomes exhibited high expression pattern differences among experimental groups, while head kidney data of 16 chromosomes were identified with high differences among experimental groups (Figure 2C,D). Moreover, in both tissues, syntenic blocks between chromosome regions with a high CGE index (expression differences) were observed, suggesting the expression of genes with duplication in the salmon genome.

Among the genes upregulated in the head kidney from fish immunized with cathepsin, haptoglobin, tyrosine kinase, and cathepsin X were mainly identified. In addition, the differential expression analysis among the genes in the CGE areas showed several genes modulated in fish immunized with cathepsin in the head kidney (487 transcripts) compared to the group immunized with peritrophin and P/C (Appendix A). The highest number of transcripts differently modulated in skin tissue was associated with the peritrophin antigen (311 transcripts), where *HSP70, serine/threonine-protein phosphatase,* and *TLR22a2* gene were significantly expressed (Appendix A).

### 3.3. Transcriptome Profile of C. rogercresseyi Female Obtained from Immunized Atlantic Salmon

The transcriptome profiling of sea lice collected from Atlantic salmon showed pronounced differences among the experimental groups (Figure 3A). Lice sampled from fish exposed to peritrophin and cathepsin were rooted in the same clade, while those exposed to fish immunized with both antigens were separately grouped. The heatmap representation exhibited two main clusters of transcripts (Figure 3A). Notably, Cluster 1 showed up-modulated genes such as vitellogenin and *cathepsin* D (Figure 3B). However, *vitellogenin* transcripts identified in Cluster 2 exhibited low expression in the cathepsin and P/C groups compared with the control and peritrophin experimental groups. Moreover, Cluster 2 was observed with high expression levels of trypsin-1 and collagenase in lice collected from all experimental groups (Figure 3B), where several transcripts were differently modulated in lice exposed to immunized fish with cathepsin (Figure 3C). The cathepsin experimental group showed the highest number of transcripts upregulated compared to those lice sampled from fish exposed to other antigens. Thus, the cathepsin antigen appears to significantly impact the sea louse transcriptome modulation.

Among the genes differently modulated in lice collected from fish immunized with peritrophin, *Putative cuticle protein*, *tyrosine-protein phosphatase*, *glutathione synthetase, trypsin-1, perithrophin-1*, and *collagenase* were mainly annotated, which were down-modulated (Table 1). Meanwhile, for the sea lice from the cathepsin groups, upregulation of *cathepsin B, vitellogenin-2,* and *ferritin* genes was observed (Table 1). Concerning the genes differentially modulated in lice exposed to fish immunized with both antigens, down-modulation of genes was associated with the secretome, such as *serine hydrolase-like protein 2, serine protease,* and *metalloendopeptidase* (Table 1).

### 3.4. Chromosome Expression of Lice Exposed to Immunized Salmon

*C. rogercresseyi* genome expression was evaluated, following a similar pipeline to the Atlantic salmon data. The Circos representation of chromosomes with a CGE index of over 60% showed a strong regulation of chromosomes 1, 2, 3, 4, 5, 7, 9, 10, 11, 12, 13, 14, 15, 16, 17, 18, 19, and 20 in female sea lice obtained from immunized fish (Figure 4A). Some chromosomes share syntenic blocks in areas with a high CGE index, for instance, Chr2 with Chr4, Chr4 with Chr5, and Chr12 with Chr2 and 15 (Figure 4A). A total of 1370 genes were extracted from the CGE areas, and the differently expressed gene analysis showed a high number of genes differently modulated (fold change ≥ |2|; FDR = 0.05) in sea lice obtained from fish immunized with peritrophin (Figure 4B). A Blast analysis indicated that the differently expressed genes in sea lice obtained from fish exposed to peritrophin included toxin Tb2, retrotransposable element R2, and Zinc finger protein. Matrix metalloproteinases were upregulated in sea lice obtained from fish immunized with cathepsin. At the same time, serine/threonine kinase was upregulated among the annotated genes of the group exposed to P/C (Figure 4B).

### 3.5. GO Annotation of Differentially Expressed Genes

The GO annotation of differently expressed genes in the immunized fish and the attached sea lice showed different biological processes (BP) modulated in response to each antigen tested. BP, associated with histone acetylation, was annotated in the head kidney tissue of fish immunized with cathepsin. I-KappaB kinase/NF-kappaB was differentially expressed in the head kidney, indicating that the process is an innate immune response during sea lice infestation (Appendix A). Meanwhile, the Atlantic salmon head kidney of the P/C group exhibited processes such as cytokine-mediate signaling pathways, defense response to bacterium, and fin regeneration, among others (Appendix A).

Related to the genes differentially expressed in Atlantic salmon skin tissue, the abundance of genes associated with annotated Molecular Function (MF) and heparin-binding in the fish group immunized with cathepsin can be associated with the abundance of heme-binding genes in sea lice recovered from those fish (Appendix A). Moreover, a high abundance of genes associated with heme-binding was observed in female sea lice obtained from fish immunized with P/C (Appendix A). In addition, the skin tissue of Atlantic salmon immunized with cathepsin showed genes associated with MF linked to the secretome, such as protein serine kinase activity and lyase activity. Furthermore, sea lice obtained from Atlantic salmon immunized with cathepsin showed a high abundance of genes associated with the secretome, such as serine-type endopeptidase activity and proteins with serine/threonine kinase activity (Appendix A). Collectively, the results suggest secretome activity in the host and ectoparasite during the infestation triggered by immunization with cathepsin.

### 3.6. Transcriptional Correlation between Host-Parasite Differently Expressed Genes

The correlation analysis of DEGs revealed an interplay among genes modulated in *S. salar* and sea lice during the infective process. For instance, genes associated with *S. salar* heme degradation such as *ferritin, hemoglobin, SOD, and catalase* exhibited a positive correlation with sea lice *trypsin 1, C-type lectin, chitinase,* and *peritrophin 1* (Figure 5). Similarly, the expression correlation among *S. salar* and sea lice DEGs shared among the experimental groups was analyzed. This analysis showed a positive correlation between sea louse genes such as *ferredoxin-like protein*, *serine-threonine protein kinase*, and *trypsin 1* with *S. salar* immune genes such as *Complement component C3* and *MHC class I alpha antigen* (Figure 6). Notably, a positive correlation between the *trypsin inhibitor* and *cathepsin gene* of *S. salar* and sea lice *trypsin*-1 was observed. It also draws attention to the correlation between *S. salar heparin-binding EGF* and sea lice *cuticle proteins* (Figure 6).

### 3.7. Vaccine Efficacy and C. rogercresseyi Morphometry

The efficacy of each vaccine prototype was evaluated after 25 dpi by counting adult lice per fish. Vaccinated fish showed lower parasite loads than the control (Figure 7A). Notably, the vaccines that included cathepsin reduced the parasite load by 57%. In contrast, those in the peritrophin group only showed 12% efficacy (Figure 7A). Phenotypic alterations of *C. rogercresseyi* from vaccinated fish were observed (Appendix A). Females of *Caligus* exposed to fish vaccinated with cathepsin had anomalies of the genital segment and deformity of oocytes in the egg string (Appendix A). Sea lice vaccinated with peritrophin had a mean of 80 eggs per female, even higher than 69 in the control group (Figure 7B). Non-significant differences were observed in the geometric morphometry of the sea lice obtained from immunized fish. The first two canonical variables explained just 8.3% of the change in form (Figure 7C). The PCA analysis also showed mean differences in the form of sea lice dependent on the vaccines (Figure 7D).

## 4. Discussion

The marine ectoparasite, known as the sea louse, strongly impacts the salmon aquaculture industry worldwide. Different strategies have been developed to control this pathogen; however, there is still no effective method to cope the sea lice disease. Furthermore, the vaccine effects have been observed from the host response mechanism, but little is known about the parasite’s response to facing an immunized host. This study uses a transcriptomic approach to explore the effects of host–parasite interaction in response to sea lice vaccine prototypes formulated with Cathepsin and Peritrophin recombinant proteins. We report transcriptomic profile changes triggered in Atlantic salmon immunized with these antigens at 25 dpi and the effects on the sea lice molecular modulation to host response evasion. Moreover, phenotype changes in sea lice females taken from the immunized fish were described.

Previously, we reported the efficacy in reducing sea lice early stage in Atlantic salmon vaccinated with both antigens, Cathepsin and Peritrophin. Notably, we reported a 52% sea lice burden reduction at 7dpi in fish vaccinated with both antigens [29]. In addition, an improvement in the host’s inflammatory response in immunized fish was observed [29]. Similarly, in this study, fish immunized with both antigens showed the highest efficacy in sea lice burden reduction (57%). In addition, fish vaccinated with cathepsin showed similar efficacy. Moreover, sea lice females obtained from immunized fish showed an egg string deformity, suggesting putative effects in sea lice reproduction. These findings are congruent with previous observations reported in female sea lice collected from fish vaccinated with the IPath vaccine [17]. Interestingly, the tick *Hyalomma asiaticum* exposed to rabbits vaccinated with cathepsin L-like proteins identified in this ectoparasite has been reported to have a molting success reduction and parasite load reduction of 55% [39]. Furthermore, a physiological alteration in embryos of the tick *Rhipicephalus microplus* and parasite load reduction has been observed due to the exposition of a host vaccinated with cathepsin [40].

It is known that sea lice parasitic-resistant salmon species, such as Coho salmon, exhibit down-modulation of genes associated with the antioxidant response and host response evasion [26]. For instance, up-modulation of the cathepsin gene has been reported in sea lice obtained from the resistant specie of Coho salmon compared with sea lice collected from Atlantic salmon, a susceptible species [26]. Similarly, a down-modulation of antioxidant genes has been observed in *C. rogercresseyi* parasitizing immunized salmon with the IPath vaccine [23]. This study observed decreased expression levels in genes such as *putative cuticle protein*, *tyrosine-protein phosphatase*, *glutathione synthetase*, and *collagenase* in the sea lice obtained from fish immunized with the Peritrophin protein. Furthermore, sea lice sampled from fish immunized with cathepsin showed increased expression of the cathepsin B gene. The findings suggest improved resistance to sea lice infestation in fish immunized with cathepsin or the P/C combination that triggers a sea lice response depression.

The present study reported high enrichment of genes associated with the immune response in head kidney samples from fish immunized with peritrophin and P/C. It has been reported that MHCII, pro-inflammatory cytokines, and Th2 response genes are upregulated during *C. rogercresseyi* infestation [24]. Moreover, an expression increase in Toll-like receptors TLR21 and TLR22 has been observed [24]. In addition, Atlantic salmon immunized with the IPath vaccine decreased the expression of TLR22, MHCII, and IL-1B [17]. However, fish increased the expression of IgM and IgT in blood cells after infection by *C. rogercresseyi* [17]. Thus, the suggested immune modulation effects in fish vaccinated with our vaccine’s prototypes improve the *S. salar* response to sea lice infestation, which is reflected in the parasite burden reduction.

*C. rogercresseyi*, during their different developmental stages, can modulate different molecular mechanisms related to the infestation process [27,41]. These include expressing host-recognition molecules such as inotropic receptors in pre-infective stages [26,27,42]. Furthermore, sea lice proteins such as trypsins, serpins, and cathepsins have been identified as highly modulated in infective stages, allowing host response evasion [42,43,44]. In this study, we used an approach based on the Dual RNA-seq approach to identify the relationship between parasite genes and the response of immunized fish. Our results showed that sea lice genes’ expression strongly correlates with the expression of immune host response genes in all groups of immunized fish. For example, the high transcription of trypsin 1 in sea lice was associated with the increased expression of immune-related genes such as protein inhibitors and vasodilators in fish hosts.

Interestingly, during *S. salar* and *C. rogercresseyi* interaction, a molecular event associated with the strategy denoting nutritional immunity has been described [25,42]. Furthermore, it has been observed that fish infested with *C. rogercresseyi* modulate genes related to heme biosynthesis, ROS response, and immune response [24,45,46]. Notably, the nutritional immunity strategy is relevant for *S. salar* response to sea lice infestation, where *S. salar* showed an expression variation of genes associated with heme biosynthesis and iron transport [25,46]. Previously, our research group reported increased expression of genes associated with biological processes such as metal ion binding in fish immunized with Cathepsin and Peritrophin during *C. rogercresseyi* early infection [29]. In the present study, vaccinated fish with the lowest sea lice burden in cathepsin and P/C groups exhibited strong overregulation of functions related to micronutrients, energetic molecules, and catalytic activity. Moreover, fish hosts and sea lice showed upregulation genes associated with heme, magnesium, calcium, zinc, and iron binding. These findings suggest competition for relevant ions between the immunized host and the ectoparasite.

## 5. Conclusions

This study evaluated the molecular response of *S. salar* immunized with three vaccine prototypes and their effects on *C. rogercresseyi* exposed to immunized fish 25 days post-infection. The transcriptional expression profiling in immunized fish and sea lice collected from those fish showed vaccine-dependent patterns. The study showed an effect on sea lice burden and phenotype, consequently expanding to immunized fish. For instance, the number of eggs per egg string was reduced in sea lice collected from fish immunized with peritrophin and the P/C. In addition, host–parasite gene interplay was reported through a Dual RNA-seq approach. Herein, nutritional and immunity-related genes were upregulated in response to the increased expression of sea lice genes associated with the infestation process, such as trypsin. Finally, this study provides relevant molecular and phenotypic knowledge associated with the effect of recombinant lice antigens, showing the potential commercial application in salmon farming to control *Caligus rogercresseyi* infestations.

## Figures and Tables

**Figure 1 vaccines-10-01875-f001:**
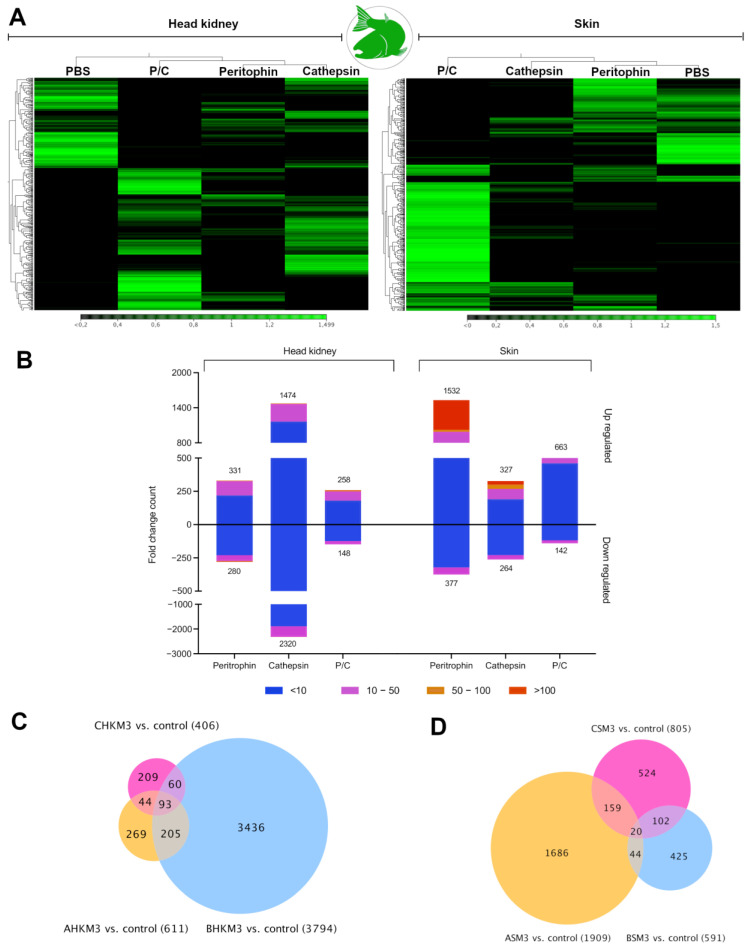
Transcriptome profiling expression of vaccinated Atlantic salmons. (**A**) Heatmaps showing expression values (TPM) in the head kidney and skin tissue. (**B**) Fold change count of up- and down-regulated contigs. The *Y*-axis was segmented for better data visualization. Venn diagram showing the distribution of contigs differentially expressed in head kidney (**C**) and skin (**D**) tissue. (Fold Change ≥ |4|; FDR = 0.05).

**Figure 2 vaccines-10-01875-f002:**
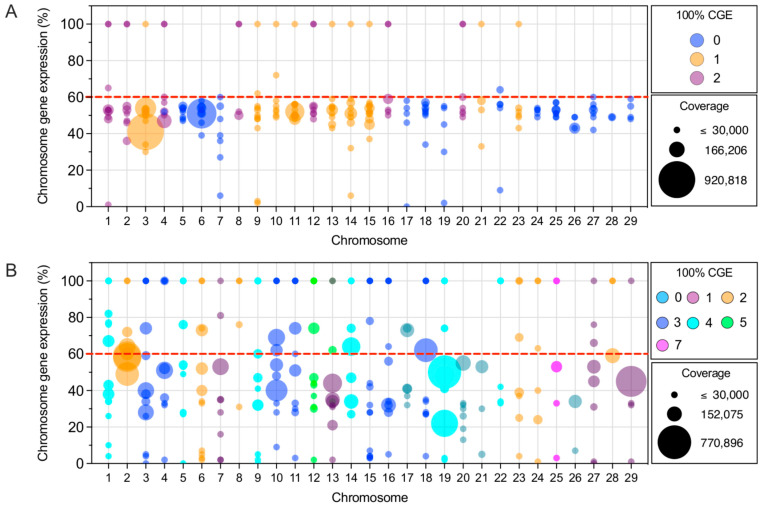
Whole-genome transcription in *S. salar* vaccinated with Peritrophin, Cathepsin, a combination of P/C, and PBS. The bubble plot shows the expression threshold analysis in vaccinated *S. salar* head kidney (**A**) and skin (**B**). The circle sizes represent chromosome coverage. Colored circles represent how many times 100% CGE is reached per chromosome. Atlantic salmon chromosome regions with CGE index variation > 60 in the head kidney (**C**) and skin (**D**). Histograms show the coverage of transcripts from different regions of the chromosomes. The heatmap in red represents the CGE index, which showed the expression variation among the four experimental groups. Syntenic blocks among *S. salar* chromosomes are also displayed.

**Figure 3 vaccines-10-01875-f003:**
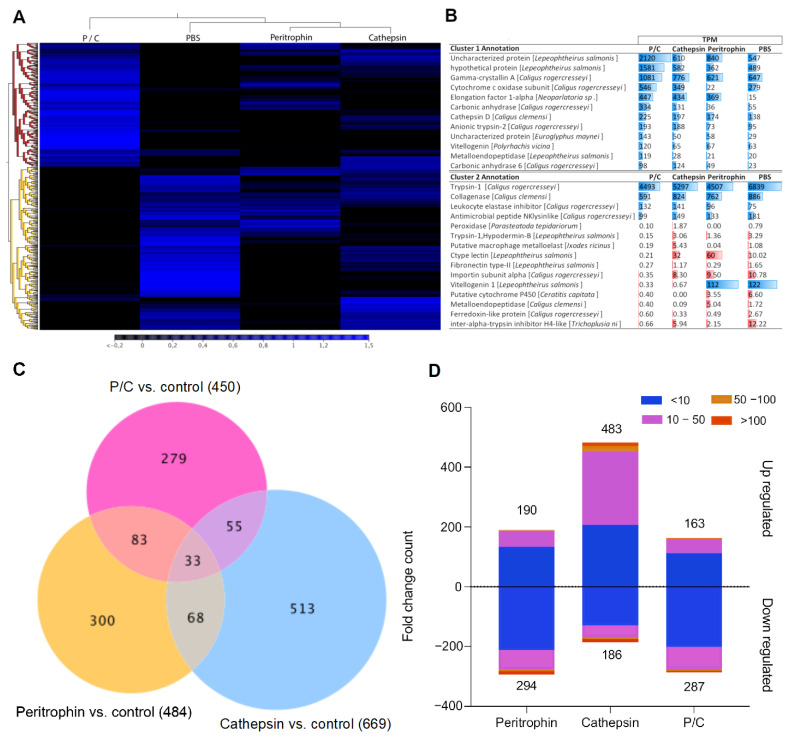
Transcriptome profiling of *C. rogercresseyi* collected from vaccinated fish. (**A**) Heatmaps showing expression values expressed in TPM. Gene clusters 1 and 2 in the RNA-Seq analysis are represented by the red and orange roots, respectively. (**B**) Annotation of contigs expressed in the clusters. Blue bars represent higher TPM values, and red bars represent lower TPM values in vaccinated fish. (**C**) Fold change count of up and downregulated contigs. (**D**) Contigs differentially expressed (Fold Change ≥ |2|; FDR = 0.05).

**Figure 4 vaccines-10-01875-f004:**
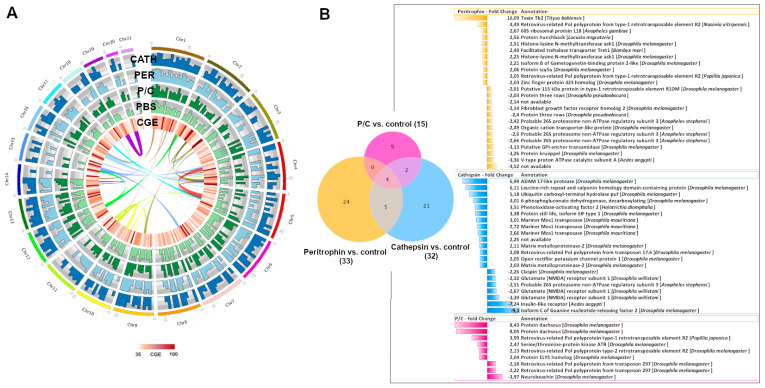
Whole-genome transcription profile of *C. rogercresseyi* collected from vaccinated fish. (**A**) Chromosome regions with CGE index variation > 60. Histograms show the coverage of transcripts from different chromosomes. Heatmap in red showed the expression variation among the four groups. (**B**) Venn diagram showing the distribution of contigs differentially expressed. Bars represent fold change values of annotated genes. (Fold Change ≥ |2|; FDR = 0.05).

**Figure 5 vaccines-10-01875-f005:**
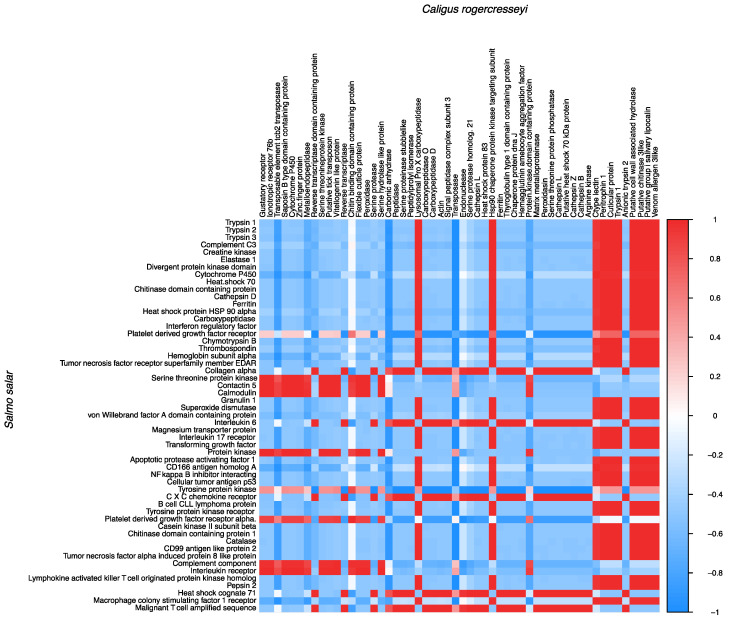
Correlation between the differentially expressed genes (DEGs) of the Peritrophin, Cathepsin, and P/C vaccine prototypes. The genes of *S. salar* and *C. rogercresseyi* are shown on the Y and X axes. The matrix shows the correlation coefficients between the TPM values of the DEGs of *S. salar* and *C. rogercresseyi*. Red and blue colors indicate significant correlations (significant positive or negative correlations, respectively).

**Figure 6 vaccines-10-01875-f006:**
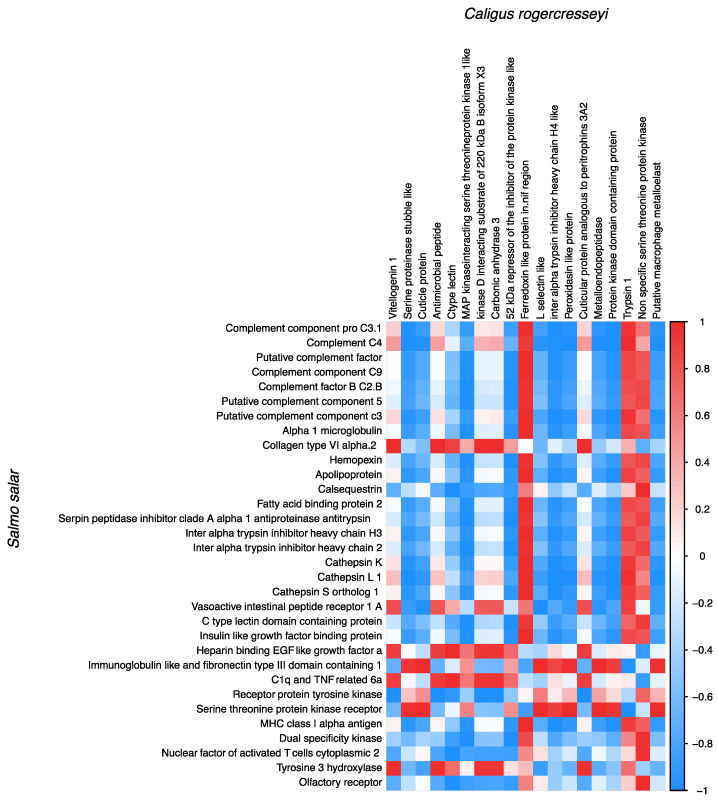
Correlation between the shared DEGs among Peritrophin, Cathepsin, and P/C vaccine prototypes. The genes of *S. salar* and *C. rogercresseyi* are shown on the Y and X axes, respectively. The matrix shows the correlation coefficients between the TPM values of the shared DEGs of *S. salar* and *C. rogercresseyi*. Red and blue colors indicate significant correlations (significant positive or negative correlations, respectively).

**Figure 7 vaccines-10-01875-f007:**
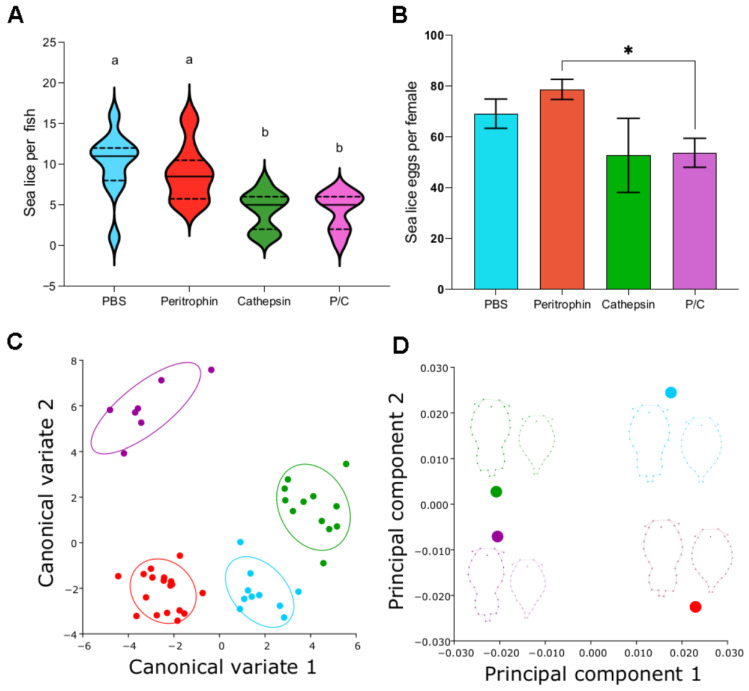
Effect of exposure of *C. rogercresseyi* to fish injected with vaccine prototypes. Treatments differentiated by colors: PBS control group in blue, peritrophin-vaccinated group in red, cathepsin-vaccinated group in green, and P/C-vaccinated group in purple. (**A**) The violin plots represent adult *C. rogercresseyi* bunder per fish in each experimental group. Lowercase letters indicate significant differences between groups at a *p*-value < 0.05. (**B**) Columns represent the median of females’ eggs per treatment, and the error bars represent the SEM. Asterisks indicate significant differences with a *p*-value < 0.05. (**C**) Morphometric analysis per canonical variables of experimental groups. (**D**) Principal components analysis of female and male morphotypes. In addition, the average morphotype figure for each group is represented.

**Table 1 vaccines-10-01875-t001:** Annotation of differentially expressed contigs in *C. rogercresseyi* exposed to vaccinated fish. Yellow bars represent downregulated contigs. Green bars represent upregulated contigs.

Peritrophin Vaccine Prototype	Cathepsin Vaccine Prototype	Peritrophin/Cathepsin Vaccine Prototype
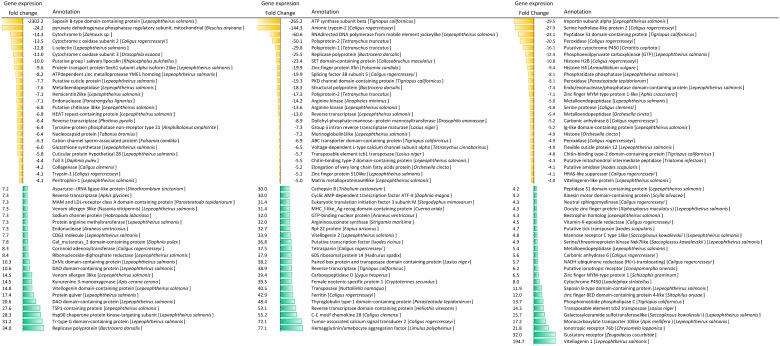		

## Data Availability

This study did not report any data.

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
