# Peer review of "Dual RNA-Seq Analysis Reveals Transcriptome Effects during the Salmon–Louse Interaction in Fish Immunized with Three Lice Vaccines"

_vaccines, 2022, doi:10.3390/vaccines10111875_

Round 1
Reviewer 1 Report
This author evaluated the immune response of S. salar immunized with three vaccine prototypes and their effects on C. rogercresseyi exposed to immunized fish 25 days post infection. Also, they applied the Dual RNA-seq analysis to demonstrate the vaccine-dependent genes in both host and pathogen, and found out a conspicuous competition for metal ions between the host and pathogen during the infestation. It’s a piece of interesting work and this manuscript will benefit the authorship of the Journal, Vaccines MPDI, however, there are still some points needed to be verified before being accepted as a publication for the Journal.
Major points:
The authors should choose some of the modulated genes in host and/or pathogen and validate them thorough qRT-PCR to confirm that the data from the RNA seqs are accurate or reliable.
Minors:
1. The abstract should be rewrite and focus on describe the major methods and results.
2. In the introduction, in line 45, it’s confusing, the ‘potassium chloride’ is a protein?
Author Response
Dear reviewer,
We do appreciate your suggestions to improve our manuscript. Please find below the responses to every question and/or comment that you have made. Furthermore, the corrections and suggested changes to the manuscript were made using “Track Changes” to facilitate its visualization. Also, the manuscript is highlighted in different color for each reviewer’s comment.
Yours Sincerely,
The corresponding author
Reviewer 1 (green highlighted)
Major points:
The authors should choose some of the modulated genes in the host and/or pathogen and validate them through qRT-PCR to confirm that the data from the RNA seqs are accurate or reliable. R. Thanks for the comments. Expression validation analysis by RT-qPCR was included and is shown in Figure S1 (lines 141-161, 201-202).
Minors:
- The abstract should be rewritten and focus on describing the major methods and results. R. Thanks for the comments. The abstract was re-writing. (lines 11 - 29)
- In the introduction, in line 45, it’s confusing, the "potassium chloride" is a protein? R. Thanks for the comment. The sentence was corrected (line 50).
Reviewer 2 Report
1. Figure 1B shows the total number of different contigs for Cathepsin/control from Head kidney is 3794 while Figure 1C shows the number is 3494 in the bracket. They are not consistent. Please revise it to the correct number. And what is the white lanes in figure 1B?
2. Figure 2A and 2B, where are the sixteen chromosomes and twenty-five chromosomes showing in the figures?
3. Regarding the fold change of different contigs in Figure 1 (Fold change more than 4) and Figure 3 (Fold change more than 2), why authors select different value for the cutoff? Please clarify the reason.
4. There are two figures showing “Figure S4” in the supplementary material, please revise it.
5. Please make the citation of figures be consistent in the main text. Like Fig. 1A, then Figure 5 should be Fig. 5, Figure 6 should be Fig. 6 in lines 308 and 315, respectively.
6. Authors did GO annotation of the DEGs, how about the pathway enrichment analysis?
Author Response
Dear reviewer,
We do appreciate your suggestions to improve our manuscript. Please find below the responses to every question and/or comment that you have made. Furthermore, the corrections and suggested changes to the manuscript were made using “Track Changes” to facilitate its visualization. Also, the manuscript is highlighted in different color for each reviewer’s comment.
Yours Sincerely,
The corresponding author
Reviewer 2 (blue highlighted)
- Figure 1B shows the total number of different contigs for Cathepsin/control from Head kidney is 3794, while Figure 1C shows the number is 3494 in the bracket. They are not consistent. Please revise it to the correct number. And what is the white lanes in figure 1B? R. Thanks for the observation. The number of transcripts in Figure 1C was checked and corrected, following your suggestion. In Figure 1B, the white line is the Y-axis segment breaks. We use three Y-axis segments for better data visualization. In response to the reviewer's concern, we included the explanation of the Y-axis segments in the figure description (line 226).
- Figure 2A and 2B, where are the sixteen chromosomes and twenty-five chromosomes showing in the figures? R. Thanks for the question. In figures 2A and 2B, on the X-axis are represented all Salmo salar genome, the 29 chromosomes. The Circos plot, in Figure 2C, 2D showed the 16 and 25 chromosomes that have the highest differences among the experimental groups, CGE index > 60%. In response to the reviewer's request, a depth explanation of these figures was included in the main text (lines 230 - 243).
- Regarding the fold change of different contigs in Figure 1 (Fold change more than 4) and Figure 3 (Fold change more than 2), why authors select different value for the cutoff? Please clarify the reason. R. Thanks for the question. We decide use to different fold change cutoff values for salmon and sea lice data because for the sea lice data genes with fold change value ³ï4ï were a reduced number of genes, which did not allow perfume a gene enrichment analysis Therefore, to improve the exploration of sea lice response to the immunized salmon we determine to use a fold change value ³ï2ï for the sea lice differential expression analysis.
- There are two figures showing “Figure S4” in the supplementary material, please revise it. R. Thanks for the comment. We check the figure number as you suggest.
- Please make the citation of figures be consistent in the main text. Like Fig. 1A, then Figure 5 should be Fig. 5, Figure 6 should be Fig. 6 in lines 308 and 315, respectively. R. Thanks for the comment. We check the figure number as you suggest (lines 348, 352, 355).
- Authors did GO annotation of the DEGs, how about the pathway enrichment analysis? R Thanks for the question. We performed KEEG pathway enrichment analyses for our data set. However, we estimate that the GO enrichment analysis allows us to explain our in better way our results. In addition, in Figure 5 we demonstrate an expression correlation between S. salar genes and sea lice genes, differently modulated in response to the different vaccines.
Round 2
Reviewer 1 Report
This manuscript is acceptable now since the authors responded to most of the questions as required.
Author Response
Dear reviewer,
We do appreciate your suggestions to improve our manuscript. Please find below the responses to every question and/or comment that you have made. Furthermore, the corrections and suggested changes to the manuscript were made using “Track Changes” to facilitate its visualization. Also, the manuscript is highlighted in different color for each reviewer’s comment.
Yours Sincerely,
The corresponding author